# Effects of a self-guided digital mental health self-help intervention for Syrian refugees in Egypt: A pragmatic randomized controlled trial

Sebastian Burchert[1]*, Mhd Salem Alkneme[1], Ammar Alsaod[1], Pim Cuijpers[2,3], Eva Heim[4], Jonas Hessling[1], Nadine Hosny[4,5], Marit Sijbrandij[2], Edith van't Hof[6], Pieter Ventevogel[7], Christine Knaevelsrud[1], on behalf of the STRENGTHS Consortium

1 Department of Education and Psychology, Division of Clinical Psychological Intervention, Freie Universität Berlin, Berlin, Germany, 2 Department of Clinical, Neuro and Developmental Psychology, WHO Collaborating Center for Research and Dissemination of Psychological Interventions, Amsterdam Public Health Research Institute, Vrije Universiteit Amsterdam, Amsterdam, the Netherlands, 3 Babeș-Bolyai University, International Institute for Psychotherapy, Cluj-Napoca, Romania, 4 Department of Psychology, University of Lausanne, Lausanne, Switzerland, 5 Department of Psychology, The American University in Cairo, New Cairo, Egypt, 6 Independent researcher, 7 Public Health Section, Division of Resilience and Solutions, United Nations High Commissioner for Refugees, Geneva, Switzerland

* s.burchert@fu-berlin.de

## Abstract

### Background

Digital mental health interventions for smartphones, such as the World Health Organization (WHO) Step-by-Step (SbS) program, are potentially scalable solutions to improve access to mental health and psychosocial support in refugee populations. Our study objective was to evaluate the effectiveness of SbS as self-guided intervention with optional message-based contact-on-demand (COD) support on reducing psychological distress, functional impairment, symptoms of posttraumatic stress disorder (PTSD), and self-identified problems in a sample of Syrian refugees residing in Egypt.

### Methods and findings

We conducted a 2-arm pragmatic randomized controlled trial. A total of 538 Syrians residing in Egypt with elevated levels of psychological distress (Kessler Psychological Distress Scale; K10 > 15) and reduced psychosocial functioning (WHODAS 2.0 > 16) were randomized into SbS + CAU (N = 266) or CAU only (N = 272). Primary outcomes were psychological distress (Hopkins Symptom Checklist 25) and impaired functioning (WHO Disability Assessment Schedule 2.0) at 3-month follow-up. Secondary outcomes were symptoms of PTSD (PTSD Checklist for DSM-5 short form, PCL-5 short) and self-identified problems (Psychological Outcomes Profiles Scale, PSYCHLOPS). Intention-to-treat (ITT) analyses showed significant but small effects of condition on psychological distress (mean difference: −0.15; 95% CI: −0.28, −0.02; p = .02) and functioning (mean difference: −2.04; 95% CI: −3.87, −0.22; p = .02) at 3-month follow-up. There were no significant differences between groups

**Data Availability Statement:** The data collected for this study involves sensitive information obtained

from Syrian refugees. Due to ethical concerns regarding the potential misuse of this data by individuals or groups with political agendas, we are unable to make it publicly available. The potential for this data to be misinterpreted or used out of context poses significant risks, including the possibility of further marginalization or stigmatization of refugee populations. In agreement with the European Commission, all members of the STRENGTHS consortium have agreed that access to the data collected under the EU-funded STRENGTHS project is instead ensured via a central repository managed by the Vrije Universiteit Amsterdam (VU). Access to this data is available upon reasonable request to the STRENGTHS consortium. Please note that access may be restricted for third parties if it conflicts with data protection laws applicable in the participating countries or with relevant EU legislation. Interested researchers can contact data steward Alex van der Jagt at apc.vander.jagt@vu.nl for inquiries and to initiate the process.

**Funding:** The STRENGTHS project was funded under Horizon 2020 – the Framework Programme for Research and Innovation (2014-2020; grant agreement 733337). The funder did not play any role in the study design, data collection and analysis, decision to publish, or preparation of the manuscript.

**Competing interests:** The authors have declared that no competing interests exist.

**Abbreviations:** CAU, care-as-usual; COD, contact-on-demand; HSCL-25, 25-item Hopkins Symptom Checklist; ITT, intention-to-treat; LMIC, lower middle-income country; MAR, missing at random; MENA, Middle East and North Africa; MHPSS, mental health and psychosocial support; MICE, Multivariate Imputation by Chained Equations; NGO, nongovernmental organization; PCL-5, Posttraumatic Stress Disorder Checklist for DSM-5; PMLD, Post Migration Living Difficulties Checklist; PSYCHLOPS, Psychological Outcomes Profiles Scale; PTSD, posttraumatic stress disorder; RCT, randomized controlled trial; SbS, Step-by-Step; SRI, Service Receipt Inventory; WHO, World Health Organization; WHODAS, WHO Disability Assessment Schedule.

on symptoms of PTSD and self-identified problems. Remission rates did not differ between conditions on any of the outcomes. COD was used by 9.4% of participants for a median of 1 contact per person. The main limitations are high intervention dropout and low utilization of COD support.

## Conclusions

The trial provides a real-world implementation case, showing small positive effects of a digital, potentially scalable and self-guided mental health intervention for Syrian refugees in Egypt in reducing psychological distress and improving overall functioning. Further user-centered adaptations are required to improve adherence and effectiveness while maintaining scalability.

## Trial registration

German Register for Clinical Studies DRKS00023505.

---

## Author summary

### Why was this study done?

- Syrians are the largest forcibly displaced population worldwide.

- Host countries like Egypt struggle to provide mental health and psychosocial support at scale for refugee populations.

- Digital self-help interventions may be a highly scalable approach to strengthening local healthcare systems.

### What did the researchers do and find?

- We tested a smartphone app offering a self-help intervention in displaced people from Syria residing in Egypt who reported elevated levels of psychological distress and problems with daily functioning.

- A total of 538 participants were randomly assigned to either use the app or to a control group receiving usual care and basic information only.

- Participants who used the app showed stronger improvements in psychological distress and daily functioning compared to the control group. However, many participants did not complete the app-based intervention and dropped out of the study early.

### What do these findings mean?

- The app was found to improve psychological distress and functioning in study participants.

- This self-help approach can be made available to many people, offering a potentially scalable solution to mental health support.

- The large number of participants who dropped out of the study indicates that the approach may only be relevant or helpful for some, and the results should be interpreted with this limitation in mind.

## 1. Introduction

Syrians are currently the largest forcibly displaced population worldwide. Since 2011, more than 5 million individuals left Syria to seek refuge in neighboring countries, including Egypt [1]. While there is growing evidence for key resilience factors among refugees [2–5], many experience increased levels of psychological distress and associated functional impairment in everyday life [6]. The accessibility and capacities of local healthcare systems in host countries impact long-term trajectories of mental health in refugee populations [7]. Postmigration living difficulties and other contextual factors further shape the types of common problems that refugees face [8,9] and need to be considered when developing interventions do address mental health issues in refugees.

Egypt is a lower middle-income country (LMIC) in northern Africa with an Arabic-speaking and predominantly Muslim population of over 110 million. In 2021, Egypt was the host country for 136,727 Syrian refugees who lived within Egyptian communities across the country [10]. Syrian refugees in Egypt exhibit elevated levels of common mental disorders including depression, anxiety, and posttraumatic stress disorder (PTSD), with high levels of comorbidity and increased levels of suicidality [11,12]. Egypt has a developed primary care system in which Syrian refugees have access to national primary care services [13]. In recent years, the country invested in improving access to mental health and psychosocial support (MHPSS) services [14]. Despite these efforts, due to the country's structural economic challenges and limited healthcare funding, MHPSS services are not integrated across the board in the Egyptian primary care system. Specialized services are available, yet, access to them remains difficult as public services are often understaffed, and higher-quality private services require out-of-pocket payment. Reaching Syrian refguees in Egypt for targeted interventions is challenging. Unlike those residing in camps or segregated communities, they are dispersed across urban and peri-urban areas throughout the country. Consequently, nongovernmental organizations (NGOs) play a crucial role in providing alternatives to refugees seeking MHPSS in Egypt [15].

The STRENGTHS project is a European Union–funded research program aimed at strengthening mental healthcare systems for Syrian refugees in key host countries, including Egypt. The project evaluated a selection of potentially scalable approaches developed by and in collaboration with the World Health Organization (WHO), covering individual, group, and digital intervention formats [16]. In pragmatic randomized controlled trials (RCTs) [17], STRENGTHS investigated these programs under real-world conditions, focusing on their effectiveness [18].

A self-guided digital intervention approach is included in STRENGTHS due to characteristics long deemed beneficial for scaling-up in hard-to-reach populations, including low access threshold, geographic and time flexibility, anonymity, and reduced fear of stigmatization [19]. There is growing evidence that digital interventions can be successfully implemented in low-resource settings [20]. For Syrian refugees, there is only a small number of studies on digital

interventions that find a range from small, nonsignificant effects [21,22] to medium effects [23] on mental health outcomes. Smartphones are of crucial importance to Syrian refugees [24] who often demonstrate high levels of technological literacy [25]. Consequently, a smartphone app–focused approach aligns well with the technology preferences of Syrians in Egypt. For instance, including offline capabilities in apps can address challenges like expensive mobile internet and poor coverage [26]. However, there is limited evidence specifically for smartphone apps as a format for digital mental health as compared to more conventional web-based programs [27,28]. Especially for refugee populations, smartphone based offers are still rare and trials have been found to struggle with recruitment and adherence [29]. These challenges can be mitigated through user-centered cultural and contextual adaptation [30,31].

Step-by-Step (SbS) is a WHO intervention for depression that was developed with a strong focus on adaptability in the areas of content, guidance, and delivery system [32]. Following a user-centered approach, the SbS content was adapted to the needs of Lebanese, Syrians, and Palestinians residing in Lebanon [33]. As part of STRENGTHS, a software platform for the delivery of SbS as a smartphone app for Syrian refugees was developed and evaluated in trials in Egypt, Germany, and Sweden [26]. Additional adaptations of SbS have been tested with Albanian [34], Chinese [35], and Filipino [36] populations. Initial studies on a guided self-help version of SbS with weekly phone contacts in Lebanon found positive treatment effects on indicators of psychological distress and functioning in Syrian refugees and Lebanese local populations [23,37,38]. In STRENGTHS, further adaptations to enhance the potential scalability of SbS were made, replacing the weekly guidance model with a self-guided, contact-on-demand (COD) model.

The main objective of the pragmatic RCT in Egypt was to evaluate the effectiveness of the adapted COD version of SbS on psychological distress, functional impairment, symptoms of PTSD, and self-identified problems in a sample of Syrian refugees residing in Egypt. In terms of refugee mental health care, and specifically digital intervention approaches, Egypt is not a well-explored setting. To our knowledge, our study is one of the first trials of this kind conducted in Egypt. It broadens the evidence base for SbS and similar digital mental health interventions.

## 2 Methods

### 2.1. Design

The study was conducted as a 2-arm RCT with preregistration in the German Register for Clinical Studies (DRKS00023505). It is reported according to the Consolidated Standards of Reporting Trials (CONSORT) [39]; see S1 CONSORT Checklist. To estimate the real-world impact of SbS, 1 study arm was given access to the 5 sessions of SbS while being free to access any other services available (care-as-usual; CAU). Depending on the individual case, CAU included no treatment, primary care, specialized mental health care, medication, or alternative healthcare approaches. The other study arm had access to CAU only and received 1 short information session. CAU was selected as the control condition to mitigate the risk of effect size inflation observed in RCTs employing a waiting-list design [40]. Given that digital mental health apps are not intended to replace other sources of mental health support, the SbS arm was not denied the use of available CAU services during SbS use to determine the incremental benefit of SbS under naturalistic conditions. Ethical approval for the study protocol (see S1 Study Protocol) was given by the Freie Universität Berlin Ethical Review Board (161/2017) and by the American University in Cairo Institutional Review Board (2020-2021-009). Deviating from the study protocol, exposure to traumatic events was not assessed and a short version of the Posttraumatic Stress Disorder Checklist for DSM-5 (PCL-5) was used in place of the full

20-item version. These adjustments responded to early participant feedback during the pilot study, which indicated that baseline assessments were overly extensive and too focused on trauma, contributing to study dropout. This paper concentrates on the results of the RCT, while the findings from the health systems analysis have been published separately [15].

## 2.2. Participants and procedure

Power calculations suggested a minimum sample size of 266 participants per group based on an anticipated effect size of 0.4 (power = 0.90, a = 0.05, two-sided) at the 3-month follow-up and considering a dropout of 50%. For this reason, a total of 532 study participants were targeted. Participants were Arabic-speaking Syrian refugees with a basic literacy level and access to the internet on an iOS or Android device or through a web browser. The app provided study information, data protection information, and a form for electronic informed consent. All steps of account creation, consent, screening, study inclusion, and randomization took place in the app. To prevent unequal group sizes, randomization was carried out via a permuted block randomization algorithm that ensured 1:1 allocation within randomly generated blocks of size 2 to 8. The block size equals the number of participants randomized within the block. To be included in the trial, participants had to show elevated levels of psychological distress (Kessler Psychological Distress Scale; K10 > 15) [41] and reduced psychosocial functioning (WHO Disability Assessment Schedule; WHODAS 2.0 > 16) [42]. Persons under the age of 18 and those who were identified with imminent risk of suicide (assessed with a self-report item on serious thoughts or a plan to end one's life) were excluded and referred to appropriate services. Participant recruitment was carried out in collaboration with Caritas Egypt, an NGO with a long track record in providing health services to refugees in the Alexandria metropolitan area. The NGO team reached out to potential participants, provided information on the study, and supported study processes onsite. Interested individuals downloaded the SbS app onto their mobile device or accessed the web version of the app through a web browser on an internet-capable device.

## 2.3. Measures

Assessments were conducted at baseline, 6 weeks after baseline (post) and 3 months after post (follow-up). Study personnel had access to information on study arm assignment for each participant through the web-based SbS study management application. To ensure that this had no impact on the data collection, all assessments were facilitated fully autonomously by the SbS app and, therefore, were not affected by potential outcome assessor bias. In the SbS app, assessments were presented as questionnaires in written format with optional prerecorded audios of all instructions, questions, and answer options. Participants received a compensation of 150 Egyptian pounds (EGP; equivalent to 9 USD) for the post and the follow-up assessments, respectively.

The primary outcomes were assessed at post and 3-month follow-up. The 25-item Hopkins Symptom Checklist (HSCL-25) [43] was used as a measure of psychological distress rated on a 1 to 4 scale (total range 25 to 100), with higher scores indicating higher psychological distress. At baseline, the reliability (Cronbach's α) was 0.94. The 12-item WHO Disability Assessment Schedule (WHODAS 2.0) [42] was used as a measure of psychosocial functioning with ratings on a 1 to 5 scale (total range of 12 to 60) across the domains of cognition, mobility, self-care, getting along, life activities, and participation. Higher scores on the WHODAS 2.0 are an indicator of lower functioning. Cronbach's α at baseline was 0.87.

Secondary outcomes were the short form of the PTSD Checklist for DSM-5 (PCL-5 short) [44], covering all diagnostic dimensions of the DSM-5 PTSD diagnosis with 9 items rated on a

0 to 4 scale (total range 0 to 36) with higher scores indicating higher symptom load. Cronbach's α for this measure was 0.90 at baseline. As an indicator of personalized intervention outcome, the Psychological Outcomes Profiles Scale (PSYCHLOPS; [45]) was administered. PSYCHLOPS consists of 2 questions on self-defined problems that participants encounter in their daily lives. Two additional questions assess functioning and general well-being in relation to these problems. The questions are rated on a 0 to 5 scale (total range 0 to 20). Higher scores on the PSYCHLOPS indicate a greater perceived burden of self-defined problems. Cronbach's α for the PSYCHLOPS was 0.83 at baseline.

Additional measures included single-item demographic questions on gender, age, education, marital status, and occupational status (Table 1). CAU service utilization was assessed with an adapted version of the Service Receipt Inventory (SRI) [46]. The SbS platform automatically tracked COD support frequency per participant. Postmigration stressors were assessed with the Post Migration Living Difficulties (PMLD) Checklist [47]. The PMLD assesses 17 stressors on a scale from 0 ("not a problem/did not happen") to 4 ("a very serious problem"). For this study, the item on difficulties learning the local language was not included due to Syrians and Egyptians' shared Arabic language background. The sum score for the remaining 16 items ranged between 0 and 64, with higher scores indicating greater exposure to postmigration stressors. Cronbach's α for the PMLD was 0.89 at baseline. Participants in the CAU arm completed a short contamination assessment on learning about or seeing the SbS session content or the SbS techniques. All measures were tested with the target population prior to starting the trial and were considered comprehensible and relevant.

## 2.4. Interventions

Participants in the SbS + CAU arm received access to an adapted version of SbS with a COD support model. In accordance with previous implementations of SbS [37], the intervention encompassed an introductory session and 5 brief content sessions, each designed to provide knowledge and skills based on established therapeutic techniques, including behavioral activation, stress management, reaching out for social support, and relapse prevention [32]. The 5 SbS sessions were unlocked sequentially, with each session's completion required before the next could be accessed. During the 6-week intervention phase, participants were free to engage with the sessions. However, to ensure adequate time for practicing the SbS techniques, there was a mandatory waiting period of 3 days between the completion of 1 session and the unlocking of the subsequent session. The intervention content was delivered in a narrative format modeled after a messenger conversation with the protagonist, a fictional former recipient of SbS, and the doctor, a fictional clinician who teaches SbS. To support identification with the narratives, participants were provided with a male or a female protagonist, each with 2 distinct background variations. These backgrounds broadly covered a married life of an older protagonist with children or an unmarried life of a younger protagonist without children and living with parents. Participants were able to adjust the protagonist's appearance reflecting common differences in cultural dress. Interactive exercises utilized smartphone-specific capabilities, including camera input, momentary mood tracking, and a planner with notifications. These features were woven into the sessions to support transfer into daily life. All SbS content was provided as text with optional audio recordings to increase accessibility. Participants in this arm could use other healthcare services (CAU) simultaneously and had access to COD support during the intervention period of 6 weeks. Our trial is the first to implement SbS using a COD guidance model, as opposed to the regular (typically weekly) guidance of previous trials via phone or messaging [23]. Guidance is crucial for scalability, given the significant costs associated with training and maintaining a support team. COD was chosen instead of scheduled

**Table 1. Demographic and baseline characteristics.**

|  | CAU (*n* = 272) | SbS + CAU (*n* = 266) | Completer[1] (*n* = 98) | Dropouts[2] (*n* = 168) | Total (*n* = 538) |
|---|---|---|---|---|---|
| M (SD) |  |  |  |  |  |
| **Age** | 33.98 (10.54) | 33.29 (11.20) | 33.52 (10.13) | 33.16 (11.81) | 33.64 (10.87) |
| **HSCL-25** | 2.39 (0.61) | 2.46 (0.61) | 2.42 (0.64) | 2.49 (0.60) | 2.42 (0.61) |
| **WHODAS** | 30.77 (8.03) | 31.94 (8.67) | 31.90 (9.23) | 31.96 (8.35) | 31.35 (8.36) |
| **PCL-5** | 14.84 (7.52) | 15.18 (7.32) | 14.50 (7.8) | 15.58 (7.02) | 15.01 (7.42) |
| **PSYCHLOPS** | 15.40 (4.52) | 15.02 (4.53) | 14.96 (4.38) | 15.06 (4.63) | 15.21 (4.52) |
| % (n) |  |  |  |  |  |
| **Female** | 69.1% (188) | 65.4% (174) | 75.5% (74) | 59.5% (100) | 67.3% (362) |
| **Marital status** |  |  |  |  |  |
| Never married | 15.8% (43) | 24.4% (65) | 16.3% (16) | 29.2% (49) | 20.1% (108) |
| Married | 70.2% (191) | 64.3% (171) | 77.6% (76) | 56.5% (95) | 67.3% (362) |
| Separated | 4.4% (12) | 3.8% (10) | 4.1% (4) | 3.6% (6) | 4.1% (22) |
| Divorced | 5.1% (14) | 3.4% (9) | 1.0% (1) | 4.8% (8) | 4.3% (23) |
| Widowed | 4.0% (11) | 2.6% (7) | 1.0% (1) | 3.6% (6) | 3.3% (18) |
| Other | 0.4% (1) | 1.5% (4) | 0% (0) | 2.4% (4) | 0.9% (5) |
| **Education[3]** |  |  |  |  |  |
| No education | 6.6% (18) | 7.5% (20) | 9.2% (9) | 6.5% (11) | 7.1% (38) |
| Primary | 35.7% (97) | 27.8% (74) | 23.5% (23) | 30.4% (51) | 31.8% (171) |
| Secondary | 42.3% (115) | 46.2% (123) | 50.0% (49) | 44.0% (74) | 44.2% (238) |
| University | 11.0% (30) | 14.3% (38) | 13.3% (13) | 14.9% (25) | 12.6% (68) |
| Technical | 2.6% (7) | 2.6% (7) | 0% (0) | 4.2% (7) | 2.6% (14) |
| Other | 1.8% (5) | 1.5% (4) | 4.1% (4) | 0% (0) | 1.7% (9) |
| **Occupation** |  |  |  |  |  |
| Paid work | 16.5% (45) | 21.4% (57) | 21.5% (21) | 21.4% (36) | 19.0% (102) |
| Self-employed | 12.5% (34) | 11.3% (30) | 10.2% (10) | 11.9% (20) | 11.9% (64) |
| Unpaid work | 0.4% (1) | 0.8% (2) | 0% (0) | 1.2% (2) | 0.6% (3) |
| Student | 5.5% (15) | 10.9% (29) | 7.1% (7) | 13.1% (22) | 8.2% (44) |
| Homemaker | 39.0% (106) | 27.4% (73) | 33.7% (33) | 23.8% (40) | 33.3% (179) |
| Retired | 0.4% (1) | 0.4% (1) | 1.0% (1) | 0% (0) | 0.4% (2) |
| Unemployed | 21.3% (58) | 21.8% (58) | 21.4% (21) | 22.0% (37) | 21.6% (116) |
| Other | 4.4% (12) | 6.0% (16) | 5.1% (5) | 6.5% (11) | 5.2% (28) |

[1]At least 4 out of 5 SbS sessions completed.

[2]Less than 4 SbS sessions completed.

[3]Highest education level started.

weekly phone calls to reduce this overhead and to increase the scalability of the approach. Intervention completion was defined as completion of a minimum of 4 out of 5 sessions because the fifth session essentially focused on repetition of previously learned techniques.

COD was provided by trained and supervised nonspecialist research assistants called "e-helpers", using the in-app messaging system. The e-helper team consisted of male Syrian Arabic-native speakers with an educational background in psychology and were refugees. Participants could at any time reach out to e-helpers for help with the intervention, app functionality, technical issues, or other topics, including referral to other services. The in-app messaging feature was data efficient and incurred no additional costs, although it required an internet connection to send and receive messages. E-helpers responded within 48 hours, which means that

the support was asynchronous. The aim of the COD was to primarily support technical use and additional needs as opposed to providing more in-depth motivational support in using the program as was the case in previous trials of SbS in Lebanon [23,37]. E-helpers used a decision tree–based expert system to provide standardized replies to common topics. Replies to new topics were coordinated with the study team before being sent. In their responses, e-helpers used a gender-neutral pseudonym when signing their messages to eliminate any potential influence of the helper's gender. Since COD was entirely optional, participants were free to complete SbS without ever reaching out to an e-helper.

CAU [48] comprised all available services that participants in Egypt had access to. In this trial, CAU was accompanied by a short information session presented in the SbS app. This session covered selected psychoeducative content from SbS session 1 but without the storytelling component and encouraged participants to seek out available services.

## 2.5. Statistical analyses

All analyses were conducted in R version 4.1.3 [49]. Primary data-analyses were conducted with the full intention-to-treat (ITT) datasets, including all participants randomized to one of the study conditions. The main analyses were conducted using linear mixed models to estimate treatment effects at the 3-month follow-up assessment (primary endpoint). Models were specified with study condition as fixed effect, a categorical variable for time as well as interaction terms between study condition and time. The baseline measurement of the outcome was included as a covariate to account for initial levels. Demographic variables (gender, age, marital status, education, occupation) and exposure to postmigration stressors were included as further covariates to account for potential confounders. A random effect for the individual participants was incorporated to model the variability in trajectories over time across participants. The models provided estimates for the effects of the study condition on changes in outcomes from baseline to each time point, as represented by the regression coefficients of the 2 condition × time interaction terms. Missing data in outcome variables were addressed through Multivariate Imputation by Chained Equations (MICE) [50]. Baseline scores, demographic variables, and exposure to postmigration stressors were included as predictors to improve imputation accuracy. The estimates across 100 imputations were aggregated using Rubin's rules [51]. To assess the robustness to deviations of the missing at random (MAR) assumption, we applied sensitivity analyses through delta adjustments [52]. The adjustments were applied to imputed scores in the intervention group to identify tipping points at which results changed.

For secondary outcomes, the same linear mixed models and multiple imputation approach were applied to estimate the effects of the study condition on symptoms of PTSD and self-defined problems. Hedges' g (g) as the indicator for effect size was calculated based on pooled multiple imputation estimates, using Rubin's rules. Per convention, effect sizes of 0.2 were considered as small, 0.5 as moderate, and 0.8 as large [53]. For all analyses, two-tailed tests were applied with $p < .05$ as an indicator of statistical significance.

In additional analyses, treatment response on the primary outcomes was further analysed for remission, for which a cutoff score ≤2.0 on the HSCL-25 at the 3-month follow-up was chosen [43,54]. Because the K10 was selected as the screening measure for its brevity, compared to the more extensive HSCL-25, analyses of remission rates were conducted solely among the subsample of participants who scored above the designated cutoff at baseline. For the WHODAS 2.0, a score ≤16 [55] was applied. The use of CAU services was analysed descriptively and included in exploratory analyses to assess the role of additional help-seeking behavior. Finally, we considered participants to be intervention noncompleters when they had

completed less than 4 sessions. These participants were compared to completers with regard to demographic variables, baseline distress (HSCL-25 and PCL-5 short), psychosocial functioning, self-reported problems, and COD support use. For self-reported problems, we conducted a qualitative analysis of the raw data in Arabic, coding all responses to the PSYCHLOPS questions: "Choose the problem that troubles you most" and "Choose another problem that troubles you" according to the problem categories developed by Drescher and colleagues [56].

## 3. Results

### 3.1. Participants

A total of 826 potential participants were screened, of which 538 were eligible for participation and completed the baseline assessment. The first participant was included in March 2021, and recruitment was completed in July 2021. The randomization algorithm allocated 266 (49.4%) participants to the intervention (SbS + CAU) arm and 272 (50.6%) to the CAU arm. The post-assessment was completed by 393 participants (73.0%) and the 3-month follow-up by 344 participants (63.9%). The intervention completion rate (4 out of 5 sessions) was 36.8%. Fig 1 provides further details on session completion and participant flow. Dropout rates in the SbS + CAU arm were high with 168 (63.2%) noncompleters. Dropouts were particularly common in the early stages, as 57 participants (21.4%) did not complete the introduction session. The COD option was used by a minority of 25 participants in the SbS + CAU arm (9.4%). COD use was defined as having initiated at least 1 contact with an e-helper by sending a message using the SbS in-app messaging system. The median number of messages sent by participants who utilized COD was 1 with a range between 1 and 18 messages.

Table 1 provides an overview of the demographic characteristics and baseline levels of psychological distress, functioning, PTSD symptoms, and self-identified problems of the participants. The average age was 33.6 years (SD = 10.9, range 18 to 71), with 67.3% female participants. The majority had started secondary or higher education (61.2%), were either in paid work or self-employed (30.9%), a homemaker (33.3%), or unemployed (21.6%), and 67.3% were married.

### 3.2. Primary outcomes

The ITT analysis showed an overall effect of time and an additional statistically significant small effect of condition on the HSCL-25 score at the post and the 3-month follow-up. Table 2 provides the results of the pooled linear mixed model analysis, which indicates significantly lower mean scores of the HSCL-25 in the SbS + CAU condition at 3-month follow-up with a mean difference of −0.15 (95% CI −0.28, −0.02). This corresponds to a small effect size of d = 0.23. Statistically significant effects at the post and follow-up time points were also found for the WHODAS 2.0 questionnaire. With a mean difference of −2.04 (95% CI −3.87, −0.22) at 3 months follow-up, the analysis showed a small effect size of d = 0.22 on this indicator of functioning at 3 months follow-up.

### 3.3. Secondary outcomes

Analyses on the PCL-5 short and the PSYCHLOPS data did not result in statistically significant results on any of the secondary outcome measures.

### 3.4. Additional and exploratory analyses

In sensitivity analyses, we applied delta adjustments to the imputed scores in the intervention group for outcomes that were statistically significant. These analyses revealed that deviations

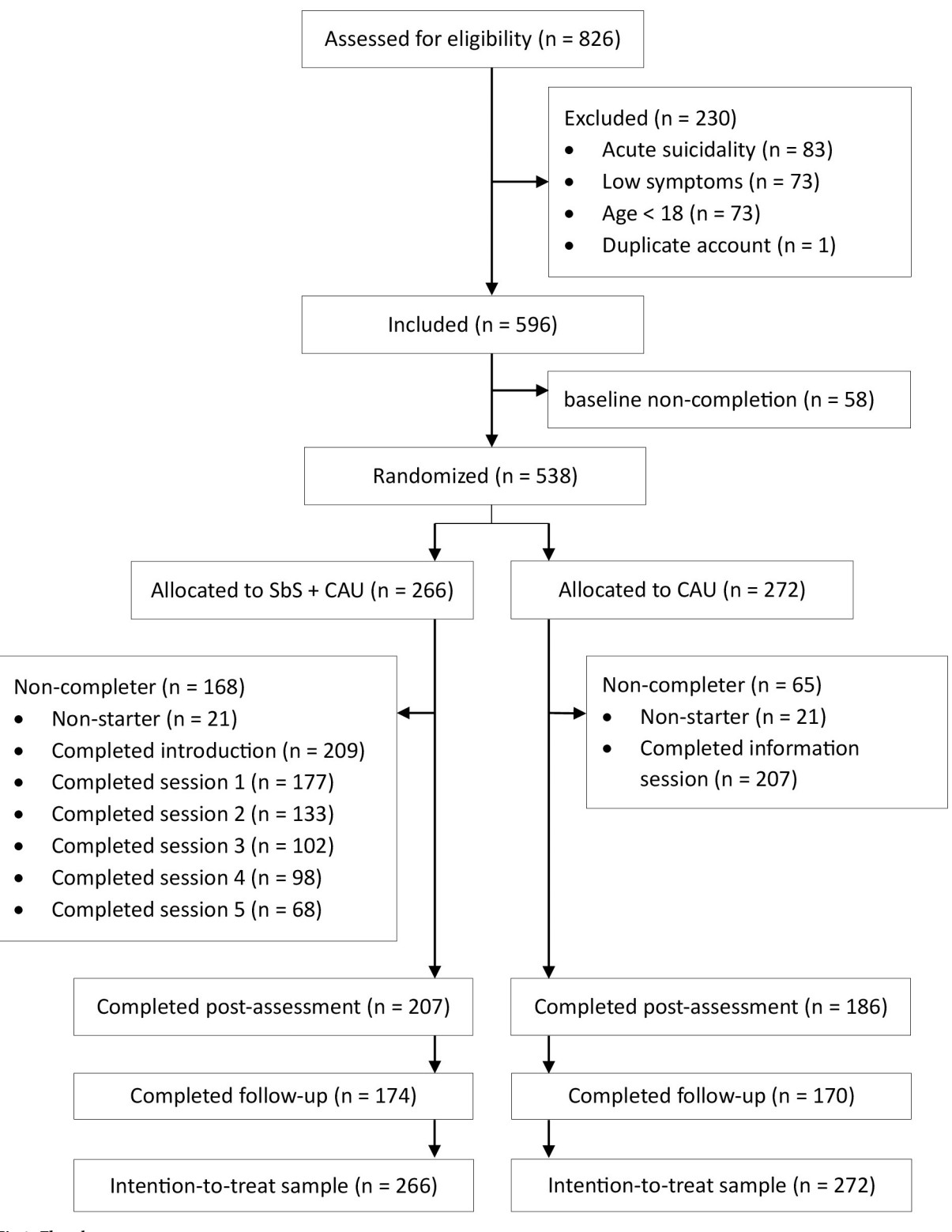

**Fig 1. Flowchart.**

**Table 2. Pooled results from linear mixed models for primary and secondary outcomes (N = 538, based on multiple imputation).**

| Outcomes | Time point | Descriptive statistics, Pooled[1] M (SD) | | | | Linear mixed model analysis[2], Pooled results[3] | | |
|---|---|---|---|---|---|---|---|---|
| | | SbS + CAU | n | CAU | n | Mean diff. (95% CI) | p-value | Effect size[4] |
| Primary | | | | | | | | |
| HSCL-25 (psych. distress) | Baseline | 2.46 (0.61) | 266 | 2.39 (0.61) | 272 | | | |
| | Post | 2.29 (0.68) | 207 | 2.36 (0.65) | 186 | −0.15 (−0.27, −0.02) | .021* | 0.23 |
| | Follow-up | 2.18 (0.67) | 174 | 2.26 (0.70) | 170 | −0.15 (−0.28, −0.02) | .022* | 0.22 |
| WHODAS (functioning) | Baseline | 31.90 (8.67) | 266 | 30.80 (8.03) | 272 | | | |
| | Post | 29.80 (8.92) | 207 | 30.90 (8.74) | 186 | −2.23 (−3.94, −0.53) | .011* | 0.25 |
| | Follow-up | 29.40 (9.11) | 174 | 30.30 (9.40) | 170 | −2.04 (−3.87, −0.22) | .023* | 0.22 |
| Secondary | | | | | | | | |
| PCL-5 (short) (PTSD symptoms) | Baseline | 15.20 (7.32) | 266 | 14.80 (7.52) | 272 | | | |
| | Post | 13.40 (7.78) | 207 | 13.30 (7.65) | 186 | −0.19 (−1.51, 1.14) | .780 | 0.03 |
| | Follow-up | 12.70 (7.98) | 174 | 13.10 (7.95) | 170 | −0.76 (−2,26, 0.74) | .320 | 0.10 |
| PSYCHLOPS (self-defined problems) | Baseline | 15.00 (4.53) | 266 | 15.40 (4.52) | 272 | | | |
| | Post | 12.80 (5.51) | 207 | 14.00 (5.18) | 186 | −0.76 (−1.65, 0.14) | .099 | 0.14 |
| | Follow-up | 12.60 (5.43) | 174 | 13.8 (5.33) | 170 | −0.85 (−1.78, 0.09) | .075 | 0.15 |

[1]Pooled descriptive statistics across all imputed datasets.

[2]As covariates the models included: baseline score, gender, age, marital status, education, occupation, and postmigration living difficulties.

[3]Treatment effects were pooled based on multiple imputations (100), assuming missing at random, using progressive mean matching (PMM).

[4]Hedges' g effect sizes were derived by combining multiple imputation estimates using Rubin's rules.

of +3% in the HSCL-25 scores (indicating higher psychological distress) and +2% in the WHODAS scores (indicating lower functioning) were sufficient to render the results not significant.

Analyses on remission rates for the HSCL-25 were conducted for the subsamples of participants who scored above the cutoff for remission at baseline assessment, which was the case in 71.4% (SbS + CAU group) and 69.9% (CAU group) of the participants. In these subsamples, full remission on the HSCL-25 was found in 20.0% (SbS + CAU group) and 15.8% (CAU group) of participants. The study conditions did not differ in their remission rates, OR = 1.33; 95% CI 0.76, 2.35; $p$ = 0.35. For the WHODAS, all participants scored above the cutoff for remission at baseline. Here, 9.8% (SbS + CAU group) and 7.1% (CAU group) were fully remitted, OR = 1.42; 95% CI 0.62, 3.38; $p$ = 0.44.

Table 3 summarizes the types of services used by study participants in both conditions throughout the intervention period of 6 weeks. More than half (58.5%) of the participants reported using at least 1 type of service during this period. Participants mainly utilized primary healthcare services by visiting a general practitioner (30.5%) or going to a health worker/nurse (23.4%). A minority reported using specialized mental healthcare services and visited a psychologist (4.1%) or a psychiatrist (3.6%), as well as taking medication for mood problems (7.1%), anxiety (11.2%), and sleep problems (14.2%). During the trial, no serious adverse events were reported to the e-helper team. Upon trial completion, analysis of the service use questionnaire indicated that 4.8% of the participants were in contact with external CAU crisis services. Hospitalization in a psychiatric hospital or a psychiatric ward in a public hospital was reported by 3 participants (0.8%). Outpatient services at a hospital were utilized by 1.5%, while 0.8% indicated a visit to an emergency room. There were no substantial differences in CAU utilization between the SbS + CAU and the CAU conditions. However, there were statistically significant differences in baseline, post, and follow-up symptom severity on the HSCL-25,

**Table 3. Service use for mental health.**

|  | SbS + CAU | CAU | Total |
|---|---|---|---|
| **Intervention period**[1] | $(n = 207)$[2] | $(n = 186)$[2] | $(n = 393)$[2] |
| 1 or more services | 56.5% (117) | 60.2% (112) | 58.3% (229) |
| 3 or more services | 25.1% (52) | 23.7% (44) | 24.4% (96) |
| 6 or more services | 3.4% (7) | 2.7% (5) | 3.1% (12) |
| Health worker/nurse | 25.6% (53) | 21.0% (39) | 23.4% (92) |
| General practitioner | 28.5% (59) | 32.8% (61) | 30.5% (120) |
| Social worker | 9.7% (20) | 14.0% (26) | 11.7% (46) |
| Physical therapist | 8.7% (18) | 7.5% (14) | 8.1% (32) |
| Home care | 16.4% (34) | 15.1% (28) | 15.8% (62) |
| Alternative medicine services | 10.1% (21) | 10.2% (19) | 10.2% (40) |
| Psychiatrist | 4.8% (10) | 2.2% (4) | 3.6% (14) |
| Psychologist | 5.3% (11) | 2.7% (5) | 4.1% (16) |
| Psychiatric nurse | 2.4% (5) | 1.1% (2) | 1.8% (7) |
| Self-help group | 0.5% (1) | 3.2% (6) | 1.8% (7) |
| Consultation center for alcohol or drugs | 0.5% (1) | 0.5% (1) | 0.5% (2) |
| Crisis service | 3.4% (7) | 6.5% (12) | 4.8% (19) |
| Psychiatric hospital | 0% (0) | 1.1% (2) | 0.5% (2) |
| Psychiatric ward in a public hospital | 0% (0) | 0.5% (1) | 0.3% (1) |
| Outpatient services at a hospital | 1.9% (4) | 1.1% (2) | 1.5% (6) |
| Emergency room | 1.0% (2) | 0.5% (1) | 0.8% (3) |
| Medication: Mood problems | 6.8% (14) | 7.5% (14) | 7.1% (28) |
| Medication: Anxiety | 11.1% (23) | 11.3% (21) | 11.2% (44) |
| Medication: Sleep problems | 13.5% (28) | 15.1% (28) | 14.2% (56) |

[1]Services used during the 6-week period between baseline and post assessment.

[2]n = number of participants who completed the post assessment.

WHODAS 2.0, and PCL-5 short. Participants who used at least 1 CAU service had higher HSCL-25 scores at baseline (M = 2.51, SD = 0.59), post (M = 2.46, SD = 0.64), and follow-up (M = 2.32, SD = 0.68). Participants who used no CAU services had lower HSCL-25 scores at baseline (M = 2.24, SD = 0.62), post (M = 2.12, SD = 0.65) and follow-up (M = 2.03, SD = 0.65). This pattern did not differ between study conditions.

Table 1 provides descriptive findings on differences between the intervention completer and intervention dropout subsamples. No differences were found for baseline distress, functioning, and demographic variables, with the exception of gender and marital status. The completer group was composed of a higher proportion of female ($\chi^2(1) = 6.30$, $p = .012$, $\varphi = .15$) and married participants ($\chi^2(1) = 2.11$, $p = .146$, $\varphi = .11$).

Table 4 summarizes the results of the qualitative analysis of self-reported problems on the PSYCHLOPS. Participants most commonly reported practical and psychological problems, as well as issues related to personal development and unmet personal needs. Comparisons between completers and noncompleters revealed notable trends. The findings indicate that participants who identified problems related to the war or the situation in Syria as their primary stressor were more likely to complete the intervention, whereas participants who reported practical problems were slightly less likely to complete SbS. While these comparisons showed no statistically significant differences, it is important to note that the study was not sufficiently powered for these analyses.

**Table 4. Categories of PSYCHLOPS self-reported problems for noncompleters and completers.**

| Problem category[1] | Problem that troubles you most | | Another problem that troubles you | |
|---|---|---|---|---|
| | Noncompleters[2] (*n* = 168) | Completers[3] (*n* = 98) | Noncompleters (*n* = 168) | Completers (*n* = 98) |
| Practical | 39.9% (67) | 34.7% (34) | 33.9% (57) | 26.5% (26) |
| Psychological | 13.7% (23) | 18.4% (18) | 10.1% (17) | 9.2% (26) |
| Interpersonal | 7.1% (12) | 3.1% (3) | 2.4% (4) | 7.1% (7) |
| Physical/psychosomatic health | 3.6% (6) | 5.1% (5) | 7.1% (12) | 3.1% (3) |
| Separation from family members | 3.6% (6) | 4.1% (4) | 2.4% (4) | 3.1% (3) |
| Related to war/home country | 1.8% (3) | 11.2% (11) | 0.6% (1) | 3.1% (3) |
| Related to family duties | 8.3% (14) | 11.2% (11) | 8.3% (14) | 10.2% (10) |
| Personal development/unmet personal needs | 10.1% (17) | 6.1% (6) | 4.8% (8) | 4.1% (4) |

[1]Problem categories according to Drescher and colleagues [56].

[2]Completed less than 4 out of 5 SbS sessions.

[3]Completed 4 or more out of 5 SbS sessions.

Further differences were identified for COD support usage. Completers sent, on average, 0.77 (SD = 2.75) messages to e-helpers compared with an average of 0.10 (SD = 0.63) messages among dropouts (t = −2.35, df = 103.05, *p*-value = 0.021). The most common communication topics included login difficulties, technical issues with internet connectivity and data downloads, and requests for referral information. Of the 25 participants who sent at least 1 message to an e-helper, 18 were intervention completers, and 7 were dropouts ($\chi^2$(1) = 47.19, $p < .001$, φ = .30). There were no differences in baseline or follow-up CAU service use between completers and dropouts.

Asked about exposure to the SbS intervention content, 51 participants (18.8%) in the CAU condition indicated having seen SbS sessions or techniques at some point during the trial, and 108 CAU participants (39.7%) reported someone telling them about SbS techniques. Excluding participants who indicated potential contamination of the CAU condition from the analyses did not change results for the primary and secondary outcomes.

## 4. Discussion

This RCT evaluated a potentially scalable self-guided digital self-help intervention for Syrian refugees in Egypt. Both groups improved over time on all outcomes measured. In addition, the SbS intervention was found to have small effects on improving our primary outcomes psychological distress and daily functioning at post-intervention and 3-month follow-up. Remission rates did not differ substantially between conditions, underlining the finding that the impact of self-guided SbS with COD support was present, but minor. There were no effects on secondary outcomes, namely, PTSD symptoms and self-defined problems. These findings must be viewed in the context of high study and intervention dropout rates.

The results add to the growing evidence base for low-threshold digital mental health offers aimed at underserved populations in low- and middle-income countries [20,57]. In the context of similar trials, this study underlines the importance of balancing key components for scalability and effectiveness when implementing digital mental health solutions. In a previous trial using SbS among Syrian refugees, which used the same version of the SbS digital app [23], significant effects were found on symptoms of depression (g = 0.61), anxiety (g = 0.41), and PTSD (g = 0.39) as well as on functioning (g = 0.45), well-being (g = 0.51), and self-defined problems (g = 0.40). In direct comparison with this trial, the effects on functioning and self-defined problems were stronger by a factor of 2 to 3 than the effects we found in the current

trial. It has to be noted that our trial focused on psychological distress instead of depression for screening and primary outcome assessment to ensure comparability with other trials in the STRENGTHS project that evaluated different intervention approaches. In the Netherlands, Syrian refugees received Problem Management Plus (PM+), a 5-session in-person intervention with a similar scope to SbS [18]. At the 3 months follow-up the authors found effects of d = 0.41 on psychological distress (HSCL-25) and d = 0.18 on daily functioning (WHODAS), whereby the former was almost twice as high compared to our trial and the latter was comparable in size. In Jordan, PM+ was offered in a group format to Syrian refugees in a camp setting [58]. This trial found 3-month effect sizes of d = .40 for the HSCL-25 depression subscale and of d = .48 for WHODAS daily functioning.

Another key difference between our trial and the previous trials on SbS in Lebanon [23,37] was the replacement of weekly scheduled e-helper contacts with a self-guided COD model. The reason to develop a COD approach was to maximize scalability. Notably, COD was only rarely used by participants, and, therefore, for most participants, the intervention was fully self-guided. This increased the overall scalability of SbS due to lower personnel requirements and enabled the inclusion of up to 100 new participants per month, as well as conducting trials in several countries at the same time. However, this adjustment to the guidance model appears to have a negative effect on the effect sizes. These findings align with meta-analytical evidence for the increased effectiveness of clinician and nonclinician guidance in digital mental health [59] and with evidence that unguided interventions are less effective than guided and face-to-face interventions [60]. However, the findings provide a missing link in the literature on digital mental health for refugee populations, specifically in regard to balancing guidance and scalability. The low utilization of the highly scalable COD offer was unexpected and could have been influenced by factors related to the following: (a) the technical implementation of the feature; (b) the manner in which the feature was introduced and explained within the app; (c) the topics and extent of e-helper support; or (d) characteristics of e-helpers. The feature's technical implementation underwent thorough testing and was designed to mirror the functionality of messaging apps, which were identified as widely familiar to most Syrians through prior research [26]. To guarantee participant awareness of the e-helper contact option, the feature was prominently introduced during study onboarding, and each participant received a welcome message from their e-helper as part of the introduction session. During the intervention, reminders of the feature were incorporated into the summary at the end of each session. When initiating contact with an e-helper, participants were required to choose from the following topics: (a) Question about SbS; (b) Motivation to continue the intervention; (c) Technical question; (d) Question about the research; or (e) Another topic. The selected topics and the overall scope of the support might not have aligned with the actual needs of the participants, as evidenced by the frequent expression of a desire to contact a mental health professional or to receive a referral for face-to-face therapy.

This aligns with another possible reason for the low uptake: participants' expectation of a more paternalistic approach to healthcare, in contrast to the self-guided and autonomous approach adopted in our trial. Paternalistic approaches remain prevalent in the Middle East and North Africa (MENA) region [61], potentially shaping expectations for a more proactive role of the e-helper in guiding participants. Finally, all Syrian e-helpers, with their own refugee experiences, possessed a deep understanding of the refugee situation. Although all helpers were male, this detail was undisclosed to participants. Participants chose their preferred grammatical gender address in Arabic, and helpers responded accordingly without revealing their own gender. A significant limitation noted by participants was that the e-helpers were not trained mental health professionals. One approach to establishing a personal connection and to build initial trust could have been to set up an initial personal or phone contact with participants, during which the messaging service could have been introduced and explained.

Intervention adherence is a crucial challenge in digital mental health and is often viewed as a larger issue in self-guided interventions due to lower engagement [62]. Although extensive user-centered adjustments were made to improve adherence prior to starting the trial [26]—including adding full audio support, streamlining content, and adding full offline capability—the dropout rate of 63.2% (i.e., completion of less than 4 sessions) was comparably high in comparison with other trials on smartphone-delivered interventions for refugee populations, where rates range between 9.2% and 80% [29]. In direct comparison with the previous trial on SbS with weekly guidance for Syrians in Lebanon [23], this trial had a slightly higher dropout rate, which is in line with previous findings on adherence in guided and unguided interventions [63]. The correlation of a lower dropout rate in participants with COD use in this trial is not clearly interpretable as a causal effect. It may as well be a result of higher initial engagement and readiness to interact with all available features of the app. This interpretation is supported by the observation that most COD users only sent 1 message in total. However, qualitative feedback from a recent study on an unguided digital intervention for PTSD, conducted with general population participants in Egypt, suggests that guidance may be perceived by participants as a factor that further enhances adherence and motivation [64].

Another critical determinant of intervention uptake and adherence is the alignment of the intervention with participants' cultural concepts of distress [65], perceived barriers to seeking help and most pressing needs. Cultural and religious beliefs may shape participants' expectations and attitude towards an offer [66]. Literature on mobile mental health acceptance among Syrian refugees underscores the importance of refraining from pathologizing responses to significant adversity, advocating instead for the utilization of terminology that resonates with the participants' experiences and cultural context [67]. Our formative research highlighted fear of stigmatization as one of the most common obstacles to seeking help. To address this, adaptations were made to ensure that SbS was culturally adequate and nonpathologizing. The adaptations were developed and evaluated in Lebanon in close exchange with experts and Syrian, Palestinian, and Lebanese community members [33]. Other engagement barriers that were mentioned by participants in the trial were (1) the high number of questionnaires as part of the baseline assessment, (2) daily responsibilities and a lack of time, (3) a lack of perceived relevance of the SbS content for the personal situation, (4) issues navigating the app, and (5) technical issues. Similar findings were reported in a recent qualitative study with SbS users from Lebanon, who reported slow internet, forgetting passwords or technical issues on older devices as reasons for low adherence, but most importantly stated that their busy lifestyle prevented them from continuing to use SbS [68].

As a pragmatic RCT, this study actively encouraged participants to seek out available CAU options in both study conditions. Analyses on CAU use revealed that service use was common with more than half of the participants using at least 1 service and 1 in 4 using 3 or more different services, including medication for mood, anxiety, or sleep. Additional exploratory analyses revealed that service use was more common in participants with higher symptom load and lower functioning. This pattern remained stable throughout the trial, indicating that those with higher levels of distress were seeking out additional support but remained more heavily burdened throughout the trial, independent of group allocation. These findings are consistent with the view that digital mental health solutions have the potential to be adopted to enhance healthcare but not to replace other existing services [69,70].

The primary limitation of the trial is the high intervention dropout rate. While not uncommon in digital intervention trials [71], low adherence reduces the meaningfulness of the results in relation to the intervention under evaluation. It is noteworthy that dropout rates were especially high at the early stages, pointing towards initial usability or user experience issues. Due to study dropout, there is a risk that the missing data were not MAR. Sensitivity analyses

revealed that the primary statistical analyses in this study are not robust against deviations from the MAR assumption. The findings of this study therefore depend on the assumption that there were no intervention group–specific factors—such as low perceived effectiveness of SbS—that led to study dropout.

Consequently, additional process evaluation research with users of SbS is warranted to understand dropout and to identify measures to increase adherence in future iterations. This includes examining self-reported problems and their role in intervention completion. It has to be noted that the SbS intervention has a strong focus on depression but was offered to participants who were screened for more general psychological distress. This may have negatively affected the match of intervention content to participant needs. Finally, adoption of the COD offer within the SbS app was unexpectedly low, which limits conclusions on this format of scalable guidance and warrants further investigation of user expectations towards this feature. Due to other available services and existing connections to the local NGO, participants also had onsite alternatives to seek additional support instead of reaching out to SbS e-helpers. Some participants also chose to reach out to the team via email, instead of using the in-app messaging system. Furthermore, a limiting factor to comparisons between this implementation of SbS in Egypt and previous studies in Lebanon are substantial contextual differences between both countries. In Lebanon, approximately 1.5 million Syrians account for roughly 25% of the population and often live in formal or informal settlements that effectively segregate local and Syrian populations [72]. Widespread poverty and barriers to acquiring legal residency further limit access to basic services, including healthcare for Syrian refugees in Lebanon [73]. In Egypt, Syrians are the largest refugee population but only account for approximately 0.1% of the population. Syrians in Egypt commonly live in urban communities [74] and experience difficult economic conditions but have access to primary, secondary, and emergency healthcare as well as financial support [75]. Differences in the implementation environment may have an impact on how an offer like SbS is perceived in the context of other available support options and on how well the narrative content components match with the actual living environments of participants. More general limitations of this trial include the potential barriers of digital literacy and smartphone access, which may restrict accessibility. Additionally, the 3-month follow-up period is too brief to ascertain long-term effects and the reliance on self-report measures could have introduced response bias. Most Syrians reside in urban areas [74] and, therefore, the recruitment strategy targeted one such area, potentially introducing bias by not reaching out to rural areas. Lastly, we found indications of possible contamination within the control group. Although this did not emerge as a confounding factor in our statistical analyses, largely due to the broad scope of the items used to assess contamination, it remains possible that participants in the intervention group shared information about or access to SbS content with others in the community.

Despite these limitations, this trial provided a real-world implementation case in a large sample of a hard-to-reach refugee population in the MENA region, namely, Syrian refugees in Egypt. It showed small positive effects on psychological distress and functioning. Even though a digital approach was chosen, working closely with a local NGO proved essential in establishing a trust base onsite within the Syrian community, resulting in high study recruitment rates while maintaining a highly scalable self-guided intervention format. The Egypt trial adds to existing findings on SbS in Lebanon, while further trials within the EU-funded STRENGTHS project on SbS in Germany and Sweden will refine them [16] and add a cost-effectiveness perspective. Importantly, our study suggests for implementers that both self-guided and helper guided versions of SbS may be beneficial and opens up the potential for different service delivery models depending on aims and available needs and resources. Potential approaches to scaling up SbS have been outlined in a previous publication [76]. Overall, empirical evidence on smartphone-based interventions is still limited, and realizing the full potential of this approach

may yet require further user-centered research to develop interactive app features that improve intervention effectiveness and adherence while maintaining scalability. Future research and implementation could also put a stronger focus on rural areas, with little access to other offers and not limited to the Syrian population. Tailoring digital intervention, such as SbS, to meet diverse contextual and individual needs presents significant challenges. These include the need to accommodate a wide variety of contexts and preferences, manage the exponential growth in content requiring quality control and maintenance, and address the technical demands of adaptive, self-guided systems. Despite these obstacles, finding elegant solutions offers substantial untapped potential.

## Supporting information

**S1 CONSORT Checklist. CONSORT standards of reporting trials.**
(PDF)

**S1 Study Protocol. Study protocol.**
(PDF)

## Acknowledgments

Step-by-Step was developed by WHO in collaboration with the National Mental Health Programme of the Ministry of Public Health in Lebanon, Freie Universität Berlin, and University of Zurich.

We thank Caritas Alexandria and the following colleagues for their support and contributions: Jinane Abi Ramia, Majdy Aldoibal, Nagwan Amin, Mina Ayoub, Kenneth Carswell, Rabih El Chammay, Nermine George, Manuel Heinrich, Mark van Ommeren, Mohamed Otefy, and Sally Sobhy.

The content of this article reflects only the authors' views, and the European Community is not liable for any use that may be made of the information contained therein.

## Author Contributions

**Conceptualization:** Sebastian Burchert, Mhd Salem Alkneme, Pim Cuijpers, Eva Heim, Jonas Hessling, Nadine Hosny, Marit Sijbrandij, Edith van't Hof, Pieter Ventevogel, Christine Knaevelsrud.

**Data curation:** Sebastian Burchert, Mhd Salem Alkneme, Jonas Hessling, Marit Sijbrandij.

**Formal analysis:** Sebastian Burchert, Mhd Salem Alkneme, Ammar Alsaod.

**Funding acquisition:** Pim Cuijpers, Marit Sijbrandij, Christine Knaevelsrud.

**Investigation:** Sebastian Burchert.

**Methodology:** Pim Cuijpers, Eva Heim, Marit Sijbrandij, Edith van't Hof, Pieter Ventevogel, Christine Knaevelsrud.

**Project administration:** Sebastian Burchert, Mhd Salem Alkneme, Ammar Alsaod, Pim Cuijpers, Jonas Hessling, Nadine Hosny, Marit Sijbrandij, Edith van't Hof, Christine Knaevelsrud.

**Resources:** Nadine Hosny, Edith van't Hof, Pieter Ventevogel.

**Software:** Sebastian Burchert, Mhd Salem Alkneme, Eva Heim, Edith van't Hof.

**Supervision:** Pim Cuijpers, Eva Heim, Marit Sijbrandij, Christine Knaevelsrud.

**Visualization:** Sebastian Burchert.

**Writing – original draft:** Sebastian Burchert.

**Writing – review & editing:** Mhd Salem Alkneme, Ammar Alsaod, Pim Cuijpers, Eva Heim, Jonas Hessling, Nadine Hosny, Marit Sijbrandij, Edith van't Hof, Pieter Ventevogel, Christine Knaevelsrud.

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
