## [Decision Letter · Decision Letter 0]

9 Feb 2024

Dear Dr. Burchert,

Thank you very much for submitting your manuscript "Effects of a self-guided digital mental health self-help intervention for Syrian refugees in Egypt: A pragmatic randomized controlled trial" (PMEDICINE-D-23-03815) for consideration at PLOS Medicine. 

As you will see, the reviewers were positive about the paper and the importance of the research topic, but they raised a number of substantial concerns about specific study details and presentation. After discussing the paper with the editorial team and an academic editor with relevant expertise, I’m pleased to offer you a chance to reply, and invite you to revise the paper in response to the reviewers’ comments. We plan to send the revised paper to some of all of the original reviewers*, and given the substantial nature of the reviewers' comments we cannot provide any guarantees at this stage regarding publication.

[LINK]

When you upload your revision, please include a point-by-point response that addresses all of the reviewer and editorial points, indicating the changes made in the manuscript and either an excerpt of the revised text or the location (eg: page and line number) where each change can be found. Please submit a clean version of the paper as the main article file and a version with changes marked should as a marked-up manuscript. Please also check the guidelines for revised papers at http://journals.plos.org/plosmedicine/s/revising-your-manuscript for any that apply to your paper.

We ask that you submit your revision by the 11th of March. However, if this deadline is not feasible, please contact me by email, and we can discuss a suitable alternative.

Don’t hesitate to contact me directly with any questions (kjanin@plos.org). If you reply directly to this message, please be sure to ‘Reply All’ so your message comes directly to my inbox.

We look forward to receiving your revised manuscript. 

Sincerely,

Katrien Janin, PhD

PLOS Medicine

plosmedicine.org

*Please note: If your article is accepted, you may have the opportunity to make the peer review history publicly available. The record will include editor decision letters (with reviews) and your responses to reviewer comments. If eligible, we will contact you to opt in or out.

Comments from the Academic Editor:

1. The PSYCHLOPS outcome measure has free text indicating the nature of the prioritised problem for that individual. Given the high drop-out, please can the authors include data on the nature of problems reported and whether certain types of problems were associated (quantitatively) with higher attrition?

2. It was not completely clear how the randomisation/allocation occured. Potential participants self-complete the consent and baseline measures and are then randomly allocated by the app? Is it waitlist control? Do they get access later if not straight away? If not, why not?

3. Please can the authors strengthen their explanation of the added value of their trial compared to the other STRENGTH trials/previous trials of this self-directed mobile-based intervention.

Comments from the reviewers:

Reviewer #1: Please see attachment

Reviewer #2: Thanks for the opportunity to read your manuscript. My role is statistical reviewer, so I have focused on the design, data, and analysis that are presented. I have put general comments first, followed by questions relevant to a specific section of the manuscript (with a page/line reference). 

The manuscript presents the results of an RCT of a smartphone app in Syrian refugees located in Egypt. There were two primary outcomes (WHODAS 2.0) and K10, operationalized as binary outcomes. The treatment group received access to a self-guided digital intervention on a smartphone initially adapted by the WHO for people in Lebanon. Randomization was 1:1 with random block size. Participants needed to be Arabic-speaking refugees in Egypt with literacy and access to a suitable smartphone, and both high K10 and lower WHODAS. Potential participants were initially contacted through an NGO providing health services to Syrian refugees. Intervention participants received access to app, a training session, and were able to contact a trained 'e-helper' on demand. Control participants received care as usual. A range of secondary outcomes was assessed (e.g. PTSD symptoms). Final assessments occurred 3 months after randomization. Main analyses were ITT, using generalized linear mixed models with a time-specific estimates of treatment effect. Rates of drop-out were relatively high, but rates similar between the two arms of the study. 

Was a separate statistical analysis plan developed for this RCT (or the overall program)? If so, is this available to view as part of the review?

Two primary outcomes were specified, but no formal adjustment for multiplicity was included. It was not clear to me from the manuscript and registered protocol if a treatment effect needed to be detected for both outcomes for the trial to be a 'success'. Was this this case? If a treatment effect for only one out of the two outcomes was needed to recommend implementation, then a formal adjustment is needed (I would recommend an alternative to Bonferonni like a Holm-Bonferroni).

Is the main analysis robust to data that are 'missing at random'? A formulation using baseline values as a covariate is (usually called 'mixed model repeated measures' in pharma trials). Given the relatively high rate of drop-out a sensitivity test for data this MAR and MNAR should be considered. There are several good options relatively easily to run with R, e.g. with MI for MAR and using delta-adjustment/tipping point in 'rbmi' backage.

Did the LMM include baseline as a covariate? The registration and protocol specify that the main outcomes/research questions are the difference in *change* in HSCL-25 and WHODAS, but in the statistical analysis section the inclusion of a baseline value (which then allows for estimates of change in the outcomes) isn't mentioned, and it looks like mean differences in outcomes between groups are presented in the results. 

P7, L193. In terms of differences in proportion, what does a Cohen's D of 0.4 translate to? 

P11, L290. I would specify the linear mixed models were used here, as K10 is often used as a binary outcome. 

Reviewer #3: Overall, the manuscript presents valuable insights into the effects of a digital mental health intervention for a vulnerable population. See specific comments below: 

1. Rationale for Combining SbS with CAU: The manuscript does not clearly articulate the rationale behind combining the Step-by-Step (SbS) intervention with Care As Usual (CAU). A more explicit explanation of this decision would provide greater insight into the study design and expected outcomes.

2. SbS Session Completion and Structure: The manuscript lacks clarity on whether participants were required to complete SbS sessions sequentially each week and if sessions were locked until the previous one was completed. Further, it's unclear how much time participants had to complete a minimum of four sessions over six weeks and whether they were restricted to one session per week or could complete multiple sessions at their discretion.

3. Completion Rates of SbS Sessions: The manuscript does not provide specific data on how many participants completed all five sessions versus the minimum four sessions. Such data would be valuable in understanding participant engagement with the intervention.

4. Analysis of Adjusted Mean Differences: The absence of adjusted mean difference analyses in the manuscript is a significant omission. Including these analyses could provide a more nuanced understanding of the intervention's effectiveness, accounting for potential confounders.

5. Dropout Rates and Attrition Analysis: The high dropout rate (63.2%) in the treatment arm is concerning. The manuscript does not sufficiently address how this attrition was handled in the analysis. Without adequately controlling for potential confounders, the reported small effect of the intervention might be overstated.

6. Trade-Off Between Scalability and Engagement: The study faces the common challenge of low engagement in app-based interventions. This issue is particularly relevant for refugees who may face barriers such as low digital literacy, education, and overwhelming life pressures in host countries. Understanding these factors in relation to engagement could provide insights for improving the intervention's effectiveness.

7. Adherence and Compliance Factors: The manuscript does not explore in depth what contributed to low adherence, compliance, and engagement with the intervention. Investigating these aspects, including the potential impact of age (mean age 33 years) on engagement, could offer valuable insights for tailoring future interventions.

8. Other Notable Limitations:

Intervention Delivery: The digital nature of the intervention might limit accessibility for participants with low digital literacy or inadequate access to technology.

Follow-Up Duration: The short follow-up period of the study leaves the long-term effects of the intervention unknown.

Self-Report Measures: The reliance on self-report measures may introduce response bias, impacting the accuracy of the data.

[LINK]

---

## [Decision Letter · Decision Letter 1]

2 Jul 2024

Dear Dr. Burchert,

Thank you very much for re-submitting your manuscript "Effects of a self-guided digital mental health self-help intervention for Syrian refugees in Egypt: A pragmatic randomized controlled trial" (PMEDICINE-D-23-03815R1) for review by PLOS Medicine.

I have discussed the paper with my colleagues and the academic editor and it was also seen again by the reviewers. I am pleased to say that provided the remaining editorial and production issues are dealt with we are planning to accept the paper for publication in the journal.

[LINK]

We look forward to receiving the revised manuscript by Jul 16 2024 11:59PM.   

Sincerely,

Katrien Janin, PhD

Senior Editor 

PLOS Medicine

plosmedicine.org

Requests from Editors:

Thank you very much for your comprehensive rebuttal letter you and the changes you have made to the manuscript. we find it much improved. For us, the following small request remain:

1. Please include the psychlops responses as you have shared in your rebuttal (response 1 to the Academic Editor - I have conferred on this with the AE and they agree). The AE notes that those who did not have a war-related problem appear much less likely to complete. The process evaluation will only be in the intervention arm whereas the psychlops data are in the full sample so it makes sense to include with this publication. These can be added to the SI if you wish so.

2. Please revises your Data availability Statement. I understand that the data are not freely available, please describe briefly the ethical, legal, or contractual restriction that prevents you from sharing it. Please also include an appropriate contact (web or email address) for inquiries (please note that this cannot be a study author).

3. Author summary 

At this stage, we ask that you include a short, non-technical Author Summary of your research to make findings accessible to a wide audience that includes both scientists and non-scientists. The authors summary should consist of 2-3 succinct bullet points under each of the following headings: 

• Why Was This Study Done? Authors should reflect on what was known about the topic before the research was published and why the research was needed. 

• What Did the Researchers Do and Find? Authors should briefly describe the study design that was used and the study’s major findings. Do include the headline numbers from the study, such as the sample size and key findings. 

• What Do These Findings Mean? Authors should reflect on the new knowledge generated by the research and the implications for practice, research, policy, or public health. Authors should also consider how the interpretation of the study’s findings may be affected by the study limitations. In the final bullet point of ‘What Do These Findings Mean?’, please describe the main limitations of the study in non-technical language. 

The Author Summary should immediately follow the Abstract in your revised manuscript. This text is subject to editorial change and should be distinct from the scientific abstract. Please see our author guidelines for more information: https://journals.plos.org/plosmedicine/s/revising-your-manuscript#loc-author-summary

4. General comment - supplementary materials: Please note that supplementary materials are not checked and will be posted as supplied by the authors. Therefore, please double check and amend it according to the relevant comments. Please cite your Supporting Information as outlined here: https://journals.plos.org/plosmedicine/s/supporting-information - Please note you may use almost any description as the item name of your supporting information as long as it contains an "S" and number. For example, “S1 Figure” and “S2 Figure,” “S1 Table” and “S2 Table.

5.General comment - social media: To help us extend the reach of your research, please provide any X (formerly known as Twitter) handle(s) that would be appropriate to tag, including your own, your co-authors’, your institution, funder, or lab. Please enter in the submission form any handles you wish to be included when we post about this paper.

Comments from Reviewers:

Reviewer #1: I've enjoyed reading through the responses which seem thorough and well-considered.

One small point, re. line 482 where it says "slightly twice as high". This language is ambiguous and (I believe) not standardised. Do you mean more than twice as high or just less than twice as high? It would be good if this could be clarified in the text.

Wishing you much success with your ongoing work.

Reviewer #2: Thanks for the revised manuscript and responses to my original review. I appreciate the clearly written responses that made the re-review much easier for me. 

The additions to the manuscript (model information + sensitivity analysis) help clarify the limitations from the high-drop out rate. Apart from one minor point (below) I have no further queries.

A minor suggestion - information about the number of randomisations in a block is usually referred to as 'block size', rather than 'block length'. 

Reviewer #3: I thank the authors for meticulously addressing all my concerns and feedback.

[LINK]

---

## [Decision Letter · Decision Letter 2]

8 Aug 2024

Dear Dr. Burchert,

Thank you very much for re-submitting your manuscript "Effects of a self-guided digital mental health self-help intervention for Syrian refugees in Egypt: A pragmatic randomized controlled trial" (PMEDICINE-D-23-03815R2) for review by PLOS Medicine.

I have discussed the paper with my colleagues and the academic editor and it was also seen again by the reviewers. I am pleased to say that provided the remaining minor editorial and production issues are dealt with we are planning to accept the paper for publication in the journal.

[LINK]

If you have any questions in the meantime, please contact me (kjanin@plos.org) or the journal staff on plosmedicine@plos.org.  

We look forward to receiving the revised manuscript by Aug 14 2024 11:59PM.   

Sincerely,

Katrien Janin, PhD

Senior Editor 

PLOS Medicine

plosmedicine.org

Requests from Editors:

Thank you very much for the revised version of your manuscript and responses to our queries.

I only have a very minor request for you at this stage before issuing the editorial acceptance of your manuscript.

The academic editor highlighted the following to us:

"Can I suggest we make the wording on the psychlops findings more conservative"

They may have inadvertently pushed you to the current wording, because they did not appreciate it was non-significant.

Suggest something along the lines of : "Comparisons between .... show no significant differences, although the analyses were under-powered." or something alike.

I have given you one week to complete the request change. However, given the very minor nature of the request, please feel free to submit sooner and it will be my pleasure to issue the acceptance of your manuscript.

Feel free to contact me at kjanin@plos.org if you have any questions about the above, or have questions in general.

Best wishes,

Katrien

Comments from Reviewers:

Reviewer #2: The revised manuscript covers the query from the last revision.

[LINK]

---

## [Editor Report · Decision Letter 3]

14 Aug 2024

Dear Dr Burchert, 

On behalf of my colleagues and the Academic Editor, I am pleased to inform you that we have agreed to publish your manuscript "Effects of a self-guided digital mental health self-help intervention for Syrian refugees in Egypt: A pragmatic randomized controlled trial" (PMEDICINE-D-23-03815R3) in PLOS Medicine.

PRESS

Sincerely, 

Katrien G. Janin, PhD 

Senior Editor 

PLOS Medicine